# Longitudinal and Lateral Control Strategies for Automatic Lane Change to Avoid Collision in Vehicle High-Speed Driving

**DOI:** 10.3390/s23115301

**Published:** 2023-06-02

**Authors:** Senlin Zhang, Xinyong Liu, Guohong Deng, Jian Ou, Echuan Yang, Shusong Yang, Tao Li

**Affiliations:** 1Key Laboratory of Advanced Manufacturing Technology for Automobile Parts, Ministry of Education, Chongqing University of Technology, Chongqing 401320, China; zhangsenlin.cqut.edu.cn@2019.cqut.edu.cn (S.Z.); lxy@stu.cqut.edu.cn (X.L.); dengguohong@cqut.edu.cn (G.D.); 2Chongqing Tsingshan Industrial, Chongqing 402760, China; yangshusong@tsingshan.com (S.Y.); litao@tsingshan.com (T.L.); 3School of Mechanical Engineering, Chongqing University of Technology, Chongqing 401320, China; yangechuan@cqut.edu.cn

**Keywords:** automobile, lane change, collision avoidance, MPC, multi objective optimization

## Abstract

The vehicle particle model was built to compare and analyze the effectiveness of three different collision avoidance methods. The results show that during vehicle high-speed emergency collision avoidance, lane change collision avoidance requires a smaller longitudinal distance than braking collision avoidance and is closer to that with a combination of lane change and braking collision avoidance. Based on the above, a double-layer control strategy is proposed to avoid collision when vehicles change lanes at high speed. The quintic polynomial is chosen as the reference path after comparing and analyzing three polynomial reference trajectories. The multiobjective optimized model predictive control is used to track the lateral displacement, and the optimization objective is to minimize the lateral position deviation, yaw rate tracking deviation, and control increment. The lower longitudinal speed tracking control strategy is to control the vehicle drive system and brake system to track the expected speed. Finally, the lane changing conditions and other speed conditions of the vehicle at 120 km/h are verified. The results show that the control strategy can track the longitudinal and lateral trajectories well and achieve effective lane change and collision avoidance.

## 1. Introduction

Over the past 20 years, the number of vehicles has been increasing [1], and the problems of environment, energy, and traffic safety have become more and more serious [2]. Automated vehicles can not only solve environmental protection and energy problems by optimizing driving behavior but also help drivers operate correctly in a complex traffic environment, so the safety of automated vehicle driving has been improved [3,4]. Changing lanes is a common driving behavior in the process of high-speed driving; how to accurately track lane-changing trajectories at high speeds has become a key issue. In order to change lanes and avoid a collision at high speed, the following three problems must be solved. Firstly, the lane-changing path is planned, then the lateral displacement tracking controller is designed, and finally, the longitudinal velocity tracking controller is designed [5,6].

In some scenarios, emergency braking cannot avoid accidents when faced with sudden obstacles. At this time, the lane change operation may avoid accidents [7]. However, there are very few products that achieve collision avoidance by changing lanes. In order to change lanes to avoid collisions, many researchers have conducted research in the trajectory planning module and trajectory tracking control module.

There are two kinds of trajectory planning: online trajectory planning and offline trajectory planning. The heuristic search method represented by A* (A-Star) algorithm has the characteristics of efficiency and flexibility [8]. Another method is a rapidly-exploration random tree, which can carry out trajectory planning in multidimensional space [9]. Sometimes, online trajectory planning is difficult to solve because the constraints conflict with each other [10]. Offline trajectory planning is fast to solve the trajectory. UC Berkeley’s Jongsang Suh et al. used the hyperbolic tangent function as a reference trajectory, and the model can limit the lateral acceleration of the vehicle to a certain extent to improve comfort [11]. Wang Zheng et al. used cubic polynomials to plan the reference trajectory of vehicles’ automatic lane-changing system [12].

Simulating the driver to control the vehicle to move along the desired trajectory through the control algorithm is the principle of trajectory tracking of the automated vehicle. Adaptive cruise control (ACC) was implemented based on fuzzy logic and controlled vehicle distance and vehicle speed, respectively, by Naranjo et al. [13]. Some researchers designed horizontal and vertical joint controllers based on sliding mode control [14,15,16]. Ji et al. proposed [17] a multiobjective model predictive control (MPC) strategy, which improved the vehicle trajectory tracking accuracy and lateral stability. Xu et al. solved the front wheel rotation angle through the MPC controller and introduced error feedback control to complete the lane change [18]. Cavanini et al. [19] proposed an MPC-based path planning algorithm that works in conjunction with supervisor logic and a commanded motion (speed/heading) following controller for solving the problem of automated vehicles crossing a road junction.

However, In the aspect of longitudinal trajectory tracking control, most studies assume that the longitudinal speed is small and constant and take the heading angle as the only control quantity, which reduces the accuracy of trajectory tracking. Therefore, there is still some research space in the aspect of transverse and longitudinal cooperative control. In order to track the collision avoidance trajectory more accurately under high-speed driving conditions, a lateral tracking controller and a longitudinal tracking controller for cooperative control were designed. Moreover, multiobjective optimization model predictive control is used for lateral displacement tracking, which improves the accuracy of vehicle trajectory tracking.

This paper has five chapters. In Section 2, the vehicle particle model is established to compare and analyze the effectiveness of vehicle collision avoidance and braking collision avoidance. In Section 3, a quintic polynomial was chosen as the reference trajectory. In Section 4, a multiobjective MPC controller is designed, which is used to track the lateral displacement with high precision. In Section 5, The longitudinal speed tracking controller is used to track the longitudinal speed of the vehicle. Simulations under typical operating conditions are performed in Section 6.

## 2. Comparison of Lane Changing and Automatic Emergency Braking Collision Avoidance

The principle of automatic emergency braking (AEB) is that the vehicle’s braking system is controlled by the electronic control unit (ECU) so that the vehicle can automatically brake to avoid colliding with pedestrians, other vehicles, etc. Additionally, the principle of automatic emergency lane changing is that the steering system and braking system of the vehicle are controlled by the ECU to avoid pedestrians and other vehicles. Figure 1 shows the automatic emergency braking and lane changing to avoid collisions.

In order to evaluate and compare the effects of steering collision avoidance and braking collision avoidance, a vehicle particle model is established and analyzed. As shown in Figure 2, the vehicle coordinate system is xoy, the ground coordinate system is XOY, and the motion equation of the vehicle in the XOY coordinate system is the following:(1)$$\ddot{X}=\frac{F_{x}\cos\varphi -F_{y}\sin\varphi }{m}$$
(2)$$\ddot{Y}=\frac{F_{x}\sin\varphi -F_{y}\cos\varphi }{m}$$
where $\ddot{X}-\ddot{Y}$ are accelerations in each axis; *m* is the mass of the whole vehicle; φ is the heading angle of the vehicle; $F_{x}$ and $F_{y}$ are the combined external forces of the tire acting on the vehicle centroid in the x and y directions under the vehicle coordinate system, respectively. There is no concern with the system inputs (control quantities) during collision avoidance, so there is no detailed derivation of $F_{x}$ and $F_{y}$ given.

From tire physical properties and Newton‘s second law [20]:(3)$${\left(\frac{\alpha_{x}}{\alpha_{x,\max}}\right)}^{2}+{\left(\frac{\alpha_{y}}{\alpha_{y,\max}}\right)}^{2}\leq 1$$
(4)$$\alpha_{x,\max}\leq \mu g$$
(5)$$\alpha_{y,\max}\leq \mu g$$
where $\alpha_{x}$ and $\alpha_{y}$ are the longitudinal and lateral accelerations of the vehicle; $\alpha_{x,\max}$ and $\alpha_{y,\max}$ are the longitudinal maximum acceleration and lateral maximum acceleration of the vehicle; μ is the road adhesion coefficient; g is Gravity acceleration. Introducing state variables $\zeta ={\left[X,Y,v_{X},v_{Y}\right]}^{T}$, the system input $u={\left[F_{x},F_{y}\right]}^{T}$, then the state equation of vehicle collision avoidance motion can be written as
(6)$$\dot{\zeta }=A_{\zeta }\zeta +B_{\zeta }u$$
(7)$$A_{\zeta }=\left[\begin{array}{cccc} 0 & 0 & 1 & 0\\ 0 & 0 & 0 & 1\\ 0 & 0 & 0 & 0\\ 0 & 0 & 0 & 0 \end{array}\right];B_{\zeta }=\left[\begin{array}{ll} 0 & 0\\ 0 & 0\\ \frac{\cos\varphi }{m} & \frac{-\sin\varphi }{m}\\ \frac{\sin\varphi }{m} & \frac{\cos\varphi }{m} \end{array}\right]$$

It is assumed that the initial state of the vehicle before collision avoidance is $X(0)=X_{0}$, $Y(0)=Y_{0}$, $v_{X}(0)=v_{X0}$, $v_{Y}(0)=v_{Y0}$, and the state of the collision avoidance end time t is $X(t)=X_{t}$, $Y(t)=Y_{t}$, $v_{X}(t)=v_{X}{}_{t}$, and $v_{Y}(t)=v_{Yt}$. Let the initial time state of the system be ${\left[0,0,v_{x},0\right]}^{T}$ so as to minimize the longitudinal collision avoidance distance required by the vehicle; that is, find the system input u to minimize $X_{t}$.

There are three kinds of collision avoidance operations, including braking collision avoidance, lane change collision avoidance and braking + lane change combined collision avoidance. The system input and constraint conditions (state constraint of the vehicle at time t) are shown in Table 1, where a is the lateral distance required by the vehicle to avoid collisions.

Using the optimization tool fminbnd in MATLAB, the minimum longitudinal collision avoidance distance required by three collision avoidance modes at different speeds is solved, respectively. The calculation results of different road adhesion coefficients are shown in Figure 3. When the vehicle is in emergency collision avoidance at high speed (v≥80km/h), the longitudinal distance required to complete lane change collision avoidance is smaller than the longitudinal distance required to complete braking collision avoidance. The higher the vehicle speed and the smaller the road adhesion coefficient, the more obvious the advantages of lane change collision avoidance.

The longitudinal distance required for combined braking and steering collision avoidance is slightly smaller than that required for collision avoidance through lane change, and the results on low-adhesion roads are closer. Therefore, when the vehicle is in high-speed emergency collision avoidance, the comprehensive performance of collision avoidance through lane change is better, which is in line with the actual situation. The following research is mainly focused on lane change collision avoidance.

## 3. Reference Path Planning for Lane Change Collision Avoidance

The common reference paths for steering collision avoidance are cubic polynomial planning, quintic polynomial planning, and hepta polynomial planning strategies. The cubic polynomial reference path expression is the following formula.
(8)$$y_{\mathrm{des}3}=\sum_{i=0}^{3}c_{i}x^{i}=c_{0}+c_{1}x^{1}+c_{2}x^{2}+c_{3}x^{3}$$
where x is the longitudinal displacement of the vehicle, $y_{\mathrm{des}}$ is the ideal lateral displacement of the vehicle, and $c_{i}$ is the fitting coefficient (similar to follow-up). The initial position of the vehicle is $\left(x_{0},y_{0}\right)$, and the end position is $\left(x_{t},y_{t}\right)$. The vehicle is traveling at a constant speed, so the vehicle’s lateral displacement $y_{0}=0$, lateral vehicle speed ${\dot{y}}_{|x=0}=0$, ${\dot{y}}_{|x=t}=0$. The lateral displacement $y_{t}=a$ of the vehicle at the end of collision avoidance and the longitudinal travel distance $x_{t}=b$. The constraint of the reference trajectory is the following formula.
(9)$$\left\{\begin{array}{l} x_{0}=0\\ y_{0}=0\\ {\dot{y}}_{|x=0}=0\\ {\dot{y}}_{|x=t}=0\\ y_{t}=a\\ x_{t}=b \end{array}\right.$$

The constraint (Equation (9)) is substituted into the cubic polynomial (Equation (8)) to obtain the reference path of the cubic polynomial:(10)$$y_{3}=3\frac{a}{b^{2}}x^{2}-2\frac{a}{b^{3}}x^{3}$$

The lateral acceleration constraints of the vehicle at the beginning and end moment of collision avoidance are taken into account in the quintic polynomial reference path, whose expression is the following formula.
(11)$$y_{des5}=\sum_{i=0}^{5}c_{i}x^{i}=c_{0}+c_{1}x^{1}+c_{2}x^{2}+c_{3}x^{3}+c_{4}x^{4}+c_{5}x^{5}$$

This reference path constraint is as follows.
(12)$$\left\{\begin{array}{l} x_{0}=0\\ y_{0}=0\\ {\dot{y}}_{|x=0}=0\\ {\ddot{y}}_{|x=0}=0\\ y_{t}=a\\ x_{t}=b\\ {\dot{y}}_{|x=t}=0\\ {\ddot{y}}_{|x=t}=0 \end{array}\right.$$
where ${\ddot{y}}_{|x=0}$ and ${\ddot{y}}_{|x=t}$ are lateral acceleration at the beginning and end of lane changing, respectively. The constraint (Equation (12)) is substituted into the quintic polynomial (Equation (11)) to obtain the reference path of the quintic polynomial:(13)$$y_{5}=10\frac{a}{b^{3}}x^{3}-15\frac{a}{b^{4}}x^{4}+6\frac{a}{b}x^{5}$$

The rate of change in lateral acceleration at the initial and end moments of the vehicle is further considered for the planning of a hepatic polynomial reference path.
(14)$$y_{des7}=\sum_{i=0}^{7}c_{i}x^{i}=c_{0}+c_{1}x^{1}+c_{2}x^{2}+c_{3}x^{3}+c_{4}x^{4}+c_{5}x^{5}+c_{6}x^{6}+c_{7}x^{7}$$
(15)$$\left\{\begin{array}{l} x_{0}=0\\ y_{0}=0\\ {\dot{y}}_{|x=0}=0\\ {\ddot{y}}_{|x=0}=0\\ y_{t}=a\\ x_{t}=b\\ {\dot{y}}_{|x=t}=0\\ {\ddot{y}}_{|x=t}=0\\ {\dddot{y}}_{|x=0}=0\\ {\dddot{y}}_{|x=t}=0 \end{array}\right.$$
where ${\dddot{y}}_{|x=0}=0$ is a lateral jerk at the state of lane changing, and ${\dddot{y}}_{|x=t}$ is a lateral jerk at the end of lane changing. The constraint (Equation (15)) is substituted into the hepatic polynomial (Equation (14)) to obtain the reference path of the hepatic polynomial:(16)$$y_{7}=35\frac{a}{b^{4}}x^{4}-84\frac{a}{b^{5}}x^{5}+70\frac{a}{b^{6}}x^{6}-20\frac{a}{b^{7}}x^{7}$$

The lateral displacements under the constraints of vehicle longitudinal speed of 72 km/h, a longitudinal distance of the obstacle ahead *b* = 50 m, and vehicle lateral offset *a* = 3. 75 m are compared under the cubic polynomial, fifth-order polynomial and hepatic polynomial planning. According to Equations (10), (13) and (16), the reference trajectories of different polynomials are drawn using Matlab, as shown in Figure 4.

The reference yaw rate and reference yaw angular acceleration for third-, fifth-, and seventh-order polynomials are compared, as shown in Figure 5 and Figure 6. Compared with the hepatic polynomial, the reference yaw rate and reference yaw angular acceleration of the quintic polynomial are relatively smaller, which means smoother lane change and better comfort. However, the seventh polynomial considers the rate of change in lateral acceleration, which reduces the longitudinal and lateral acceleration fluctuations. Moreover, its calculation process is more complicated, and the calculation results are not easy to converge, which greatly increases the time of lane change trajectory planning. In summary, quintic polynomial lane changes are relatively comfortable and efficient compared to other polynomial lane changes. Figure 7 shows the quintic polynomial reference lane change trajectory at different speeds.

## 4. Horizontal Tracking Control Strategy Based on Multiobjective Optimization

### 4.1. MPC Controller Design

The MPC controller is used to track the lateral displacement. In this paper, the vehicle dynamics model considering two degrees of freedom of lateral and yaw is shown in Figure 8.

Considering the real-time requirements of the controller, nonlinear model predictive control (NMPC) has not been adopted. The expression of the vehicle linear 2-DOF dynamic model is the following formula:(17)$$\left[\begin{array}{c} {\dot{v}}_{y}\\ \dot{\gamma } \end{array}\right]=\left[\begin{array}{cc} -\frac{C_{f}+C_{r}}{mv_{x}} & \frac{aC_{f}+bC_{r}}{mv_{x}}-v_{x}\\ \frac{aC_{f}-bC_{r}}{v_{x}I_{z}} & -\frac{a^{2}C_{f}+b^{2}C_{r}}{v_{x}I_{z}} \end{array}\right]\left[\begin{array}{c} v_{y}\\ \gamma \end{array}\right]+\left[\begin{array}{c} -\frac{C_{f}}{m}\\ \frac{bC_{r}}{I_{z}} \end{array}\right]\delta_{f}$$
where m is the mass of the subject vehicle; $C_{f}$ and $C_{r}$ are the equivalent lateral deflection stiffness of the front and rear tires, respectively; a and b are the distances from the center of mass to the front and rear axles of the vehicle, separately.$\delta_{f}$ is the front wheel steering angle; $I_{z}$ is the vehicle rotational inertia around the z axis; φ and γ are the vehicle transverse sway angle and transverse sway angular velocity, respectively.

The vehicle’s lateral displacement is the key quantity for trajectory tracking control, and its derivative ${\dot{e}}_{y}$ is defined as the following formula [11].
(18)$${\dot{e}}_{y}=v_{y}\cos\varphi +v_{x}\sin\varphi$$

Since the transverse pendulum angle is small, it can be considered that sinφ=φ and cosφ=1. Formula (2) can be simplified as follows.

This controller’s state volume is $x_{d}={\left[\begin{array}{cccc} v_{y} & \varphi & \gamma & e_{y} \end{array}\right]}^{T}$, the input is $u=\left[\delta_{f}\right]$, the lateral displacement $e_{y}$ indicates the trajectory tracking accuracy, and the yaw rate γ indicates lateral stability. The performance index of the controller is $y_{d}={\left[\begin{array}{cc} e_{y} & \gamma \end{array}\right]}^{T}$, and the prediction model is the following formula.
(19)$$\left\{\begin{array}{l} {\dot{x}}_{d}=A_{d}x_{d}+B_{d}u\\ y_{d}=C_{c}x_{d} \end{array}\right.$$

The coefficients are
$$A_{d}=\left[\begin{array}{cccc} -\frac{C_{f}+C_{r}}{mv_{x}} & 0 & \frac{aC_{f}+bC_{r}}{mv_{x}}-v_{x} & 0\\ 0 & 0 & 1 & 0\\ \frac{aC_{f}-bC_{r}}{v_{x}I_{z}} & 0 & -\frac{a^{2}C_{f}+b^{2}C_{r}}{v_{x}I_{z}} & 0\\ 1 & v_{x} & 0 & 0 \end{array}\right]$$
$$B_{d}=\left[\begin{array}{c} -\frac{C_{f}}{m}\\ 0\\ \frac{aC_{f}}{I_{z}}\\ 0 \end{array}\right];C_{c}=\left[\begin{array}{llll} 0 & 0 & 0 & 1\\ 0 & 0 & 1 & 0 \end{array}\right]$$

The prediction model is discretized into the following formula.
(20)$$\left\{\begin{array}{l} {\dot{x}}_{c}=A_{c}x_{c}+B_{c}u\\ y_{c}=C_{c}x_{c} \end{array}\right.$$

The coefficients are $A_{c}=e^{A_{d}\Delta t};B_{c}=\int e^{A_{d}\tau }B_{d}d\tau$, τ is the integration factor.

The prediction step is $N_{p}$, and the control step is $N_{c}$. It is assumed that at time k, the system prediction output vector $Y_{c}$ and the control output vector ΔU within the prediction step $N_{p}$ are
(21)$$\left\{\begin{array}{l} Y_{c}={\left[y_{c}\left(k+1|k\right)\cdots y_{c}\left(k+N_{p}|k\right)\right]}^{T}\\ \Delta U={\left[\Delta u\left(k|k\right)\cdots \Delta u\left(k+N_{c}-1|k\right)\right]}^{T} \end{array}\right.$$

The system state volume can be expressed as
(22)$$\left\{\begin{array}{l} x_{c}(k+1|k)=A_{c}x(k)+B_{c}\Delta u(k)\\ \begin{array}{l} x_{c}(k+2|k)=A_{c}^{2}x(k)+A_{c}B_{c}\Delta u(k)+\\ B_{c}\Delta u(k+1) \end{array}\\ \vdots \\ \begin{array}{l} x_{c}(k+N_{p}|k)=A_{c}^{N_{p}}x(k)+A_{c}^{N_{p}-1}B_{c}\Delta u(k)\\ +A_{c}^{N_{p}-2}B_{c}\Delta u(k)+A_{c}^{N_{p}-N_{c}}B_{c}\Delta u(k+N_{c}-1) \end{array} \end{array}\right.$$

The output expression to the prediction step $N_{p}$ is
(23)Y=Fx(k)+ΦΔU
$$\Phi =\left[\begin{array}{cccc} C_{c}B_{c} & 0 & \cdots & 0\\ C_{c}A_{c}B_{c} & C_{c}B_{c} & \cdots & 0\\ \vdots & & & \vdots \\ C_{c}A_{c}^{N_{p}-1}B_{c} & C_{c}A_{c}^{N_{p}-2}B_{c} & \cdots & C_{c}A_{c}^{N_{p}-N_{c}}B_{c} \end{array}\right]$$
$$F=\left[\begin{array}{c} C_{c}A_{c}\\ C_{c}A_{c}^{2}\\ \vdots \\ C_{c}A_{c}^{N_{p}} \end{array}\right]$$

In order to improve the tracking accuracy, it is necessary to design the objective function and define the objective quadratic function [21]:(24)$$\begin{array}{l} J=k_{q}\sum_{i=1}^{N_{p}}{\left\|e_{y,\mathrm{ref}}(k+i|k)-e_{y,\mathrm{real}}(k+i|k)\right\|}^{2}+\\ k_{p}\sum_{i=1}^{N_{p}}{\left\|\gamma_{\mathrm{ref}}(k+i|k)-\gamma_{\mathrm{real}}(k+i|k)\right\|}^{2}+\\ k_{r}\sum_{j=0}^{N_{c}-1}{\left\|\Delta U(k+j|k)\right\|}^{2} \end{array}$$

ΔU(k+j|k) is the control increment, $e_{y,\mathrm{ref}}(k+i|k)$ is the expected lateral displacement, and $e_{y,\mathrm{real}}(k+i|k)$ is the actual lateral displacement; $\gamma_{\mathrm{ref}}(k+i|k)$ is the desired yaw rate and $\gamma_{\mathrm{real}}(k+i|k)$ is the actual yaw rate; $k_{q}$ is the weight coefficient of trajectory tracking accuracy, $k_{p}$ is the weight coefficient of lateral stability, and $k_{r}$ is the weight coefficient of control increment.

Limiting the volume of control and its incremental and performance indicators to a certain range [17]:(25)$$\left\{\begin{array}{l} \delta_{\min}\leq \delta \leq \delta_{\max}\\ \Delta \delta_{\min}\leq \Delta \delta \leq \Delta \delta_{\max}\\ e_{y,\min}\leq e_{y}\leq e_{y,\max}\\ \gamma_{\min}\leq \gamma \leq \gamma_{\max} \end{array}\right.$$

Solving the optimal front wheel steering angle by quadratic programming [22], the solution vector x in the control step is $\Delta U^{*}(k)$.
(26)$$\Delta U^{*}(k)={\left[\begin{array}{cccc} \Delta u_{k}{}^{*} & \Delta u_{k+1}{}^{*} & \cdots & \Delta u_{k+N_{c}-1}{}^{*} \end{array}\right]}^{T}$$

The optimal front-wheel steering angle of the vehicle is
(27)$$\delta_{f}(k)=u(k)+u(k-1)+\Delta u_{k}^{*}$$

After inputting the optimal front wheel steering angle, the vehicle can obtain the vehicle state at time *k*, then enter the next control cycle and repeat the above steps to track the lateral trajectory.

### 4.2. Controller Tuning

The choice of controller parameters and their effect on control performance are covered in detail in this section.

The sampling time of the controller is 0.02 s, which is determined by the cycle update time of the ECU of the simulated vehicle. This guarantees that the controller has sufficient reference data to address the optimization issue. In addition, the sampling time should not be too long. Otherwise, the controller is unable to react quickly when interference occurs. The weight of the calculations will, however, grow if the sampling period is too short. The controller’s predictive step size is set to 30, which can ensure the accuracy and real-time performance of trajectory tracking. The predicted step size is too large while the vehicle speed is constant, which reduces controller sensitivity and trajectory tracking accuracy and wastes computation power. The controller will be too sensitive and cause vehicle instability and trajectory tracking failure if the predicted step size is too small. The tracking effect of the controller is not greatly affected by the control step size of the controller, but it should not be excessive. Generally speaking, we set the control step size to be between 10% and 20% of the predicted step size. Various weight coefficient choices will result in various control outcomes. For instance, raising $k_{q}$ (the weight coefficient of trajectory tracking accuracy) will improve trajectory tracking precision. Finally, the average time for one full control loop is about 10 ms, which meets the requirement of real-time performance.

## 5. Longitudinal Velocity Tracking Control Strategy

The longitudinal velocity of the vehicle is variable in the vehicle body coordinate system. As shown in Figure 9, when the vehicle speed is 72 km/h, the vehicle longitudinal speed in the vehicle body coordinate system can be obtained by adjusting the lane change completion time $t_{f}$. Therefore, it is not accurate enough to achieve only lateral trajectory tracking, and it is necessary to have the controller designed for longitudinal speed tracking.

### 5.1. Longitudinal Acceleration Decision and Driving Mode Arbitration

The purpose of the longitudinal acceleration decision is to determine the longitudinal acceleration to be used for closed-loop control based on the desired longitudinal velocity and the actual longitudinal velocity. The longitudinal acceleration decision equation is derived using the principle of preview longitudinal velocity in CarSim.
(28)$$a_{xdes}=K_{p}V_{xe}(t)+K_{I}\int_{0}^{T}V_{xe}(t)+K_{D}\frac{dV_{xe}(t)}{dt}$$
(29)$$V_{xe}(t)=v_{xdes}(t)-v_{xveh}(t)$$

In the formula, $v_{xdes}$ is the desired longitudinal speed, $v_{xveh}$ is the longitudinal vehicle speed at the current moment, T is the preview time, and $a_{xdes}$ is the desired longitudinal acceleration. $K_{p}$,$K_{I}$ and $K_{D}$ are the adjustment coefficients, respectively.

Next, the vehicle’s longitudinal driving mode needs to be arbitrated to decide whether to adopt acceleration, deceleration, or idle mode to follow the desired longitudinal speed. The switching logic of braking, driving, and idle modes of the vehicle is shown in Figure 10 below.

### 5.2. Vehicle Inverse Longitudinal Dynamics Model

After obtaining the arbitration results for the longitudinal driving mode, a vehicle inverse longitudinal dynamics model needs to be developed, and then the desired throttle opening in drive mode or the desired braking pressure in braking mode is obtained based on the desired longitudinal acceleration. The simplified structure of the inverse longitudinal dynamics model of the vehicle is shown in Figure 11.

The longitudinal kinematic equations of the vehicle are as follows:(30)$$ma_{xdes}=F_{t}-F_{b}-F_{f}(v)$$
where $F_{t}$ represents the driving force of the vehicle, $F_{b}$ denotes the braking force of the vehicle, and $F_{f}(v)$ denotes the sum of rolling resistance and wind resistance to the vehicle.
(31)$$F_{f}(v)=C_{D}A\frac{\rho_{a}v^{2}}{2}+mgf$$
where $C_{D}$ is the air resistance coefficient, A is the windward area of the vehicle, $\rho_{a}$ is the air density, and f is the rolling resistance coefficient. Without considering the elastic deformation of the tire, the driving force is calculated as follows:(32)$$F_{t}=\frac{R_{g}R_{m}\eta_{t}T_{e}\tau }{r_{r}}\left(\frac{\omega_{t}}{\omega_{e}}\right)$$

$r_{r}$ is the rolling radius of the tire, $\eta_{t}$ is the mechanical efficiency of the vehicle transmission system, $T_{e}$ is the output torque of the engine, $\omega_{t}$ is the turbine speed of the torque converter, τ denotes the variable speed ratio characteristic of the torque converter, $R_{g}$ and $R_{m}$ are the speed ratios of the transmission and the main gearbox, respectively, and $\omega_{e}$ is the speed of the engine.
(33)$$K_{d}=\frac{R_{g}R_{m}\eta_{t}\tau }{r_{r}}\left(\frac{\omega_{t}}{\omega_{e}}\right)=\frac{R_{g}R_{m}\eta_{t}\tau }{r_{r}}\left(\frac{vR_{g}R_{m}}{r_{r}\omega_{e}}\right)$$

Substituting the above equation into the driving equation yields the following:(34)$$F_{t}=K_{d}T_{e}$$

According to the formula, the expression for the desired engine output torque is obtained:(35)$$T_{des}=\frac{ma_{xdes}+F_{f}(v)}{K_{d}}$$

The inverse engine model can be obtained by combining the engine MAP diagram as follows.
(36)$$\alpha_{des}=\mathrm{MAP}(T_{des},\omega_{e})$$

Figure 12 shows the engine inverse MAP diagram.

From the previous formula, we can directly obtain the expected braking pressure [23]:(37)$$P_{des}=\frac{\left|ma_{xdes}+C_{D}A\frac{\rho_{a}v^{2}}{2}+mgf\right|}{K_{b}}$$
where $P_{des}$ is the desired braking pressure, and $K_{b}$ is the ratio between braking force and braking pressure.

## 6. Simulation

In order to verify the control effect of the vertical and horizontal joint control strategy, MATLAB and CarSim are used. The vehicle model is built in CarSim, and the control strategy is built in MATLAB. The emulation computer’s central processing unit (CPU) is AMD Ryzen 5 5600 H with Radeon Graphics 3.30 GHz, and its graphic processing unit (GPU) is an NVIDIA GeForce RTX 3050 Laptop GPU. Distance sensing is out of the scope of this work, and for the purpose of the experiments, an ideal distance sensor with zero lag is considered to be on board the automated vehicle. The simulation conditions are divided into two types: the first is the emergency lane change and collision avoidance of vehicles in a high-speed environment, and the second is the lane change in vehicles at different speeds. The quintic polynomial as the reference trajectory of lane change, longitudinal and lateral trajectory tracking control strategies are combined. Table 2 shows some of the vehicle parameters.

### 6.1. Double Lane Change Condition

Assuming that the vehicle is traveling at a speed of 120 km/h when t = 0, there is a stationary obstacle 90 m in front of the subject vehicle, and the subject vehicle must change lanes. After changing lanes, the vehicle keeps driving in a straight line. Then, another obstacle vehicle is found, the subject vehicle changes lanes again, and the subject vehicle keeps driving in a straight line after changing lanes. The collision avoidance scene is shown in Figure 13.

Figure 14 shows the response of the subject vehicle’s lateral displacement and longitudinal displacement tracking error. It can be seen that the actual trajectory of the controlled vehicle matches the reference path when driving in a straight line, but there is a small gap between the actual trajectory and the reference trajectory when the vehicle automatically changes lanes.

The response of vehicle automatic emergency lane change collision avoidance is shown in Figure 15, Figure 16 and Figure 17. As can be seen from Figure 16, the yaw rate of the subject vehicle does not fluctuate significantly, indicating that the controller responds quickly. Due to the significant coupling effect of the vertical and horizontal controller, the time delay caused by the nonlinear steering system is greatly reduced.

As can be seen from Figure 17, the subject vehicle’s longitudinal speed is basically consistent with the reference longitudinal speed. In the whole process, the maximum error value of longitudinal tracking speed is 0.43 km/h.

### 6.2. Quintic Polynomial Tracking Condition

The effect of the trajectory tracking control strategy at different vehicle speeds and the rationality of the controller parameter selection should be verified. The road adhesion coefficient is 0.7, and the speed is 72 km/h, 90 km/h, and 120 km/h, respectively. The vehicle travels 40 m on a horizontal road and then changes lanes. The quintic polynomial reference trajectories at different vehicle speeds are shown in Figure 18.

From Figure 19, it can be seen that the tracking effect will change with the speed change and the control accuracy will decrease when the vehicle speed is high. When the vehicle speed is 72 km/h, the coincidence degree between the actual trajectory and the reference trajectory is the highest; when the vehicle speed is 90 km/h, the coincidence degree between the actual trajectory and the reference trajectory is high. In the late stage of the lane change, the actual trajectory deviated slightly from the desired trajectory, and the lane change was slightly advanced; when the vehicle speed is 120 km/h, there is a certain deviation between the actual trajectory and the reference trajectory, which is more obvious in the later stage of the lane change. The lane change completion time is somewhat earlier. Overall, the designed controller is responsive and can quickly track the target trajectory without significant overshoot at the end of the lane change trajectory. Although, with the increase in vehicle speed, the lane change is completed slightly earlier, and there is a certain error between the actual trajectory and the reference trajectory, which is reasonable and within the acceptable range.

It can be seen from Figure 20, Figure 21 and Figure 22 that there are no cusps or jitters in the dynamic parameters during the whole process. There is no mutation in the yaw rate and lateral acceleration of the mass center; the peak values are within the safety range, indicating that the vehicle does not have instability phenomena such as side slip, roll, and rollover.

In the design of a lateral displacement tracking control strategy, not only the computational efficiency is considered but also the objective function, which is optimized by multiobjective optimization to achieve accurate tracking. The inverse longitudinal dynamics model is classical and easy to implement in longitudinal velocity tracking control strategy design.

## 7. Discussion

In order to demonstrate the advantages of the designed longitudinal and transverse joint control strategy, the effect of tracking the double lane change trajectory at different vehicle speeds compared to linear quadratic regulator (LQR) control is shown in Figure 23. LQR is a control technique used for linear systems. It aims to design a feedback controller that minimizes a cost function representing system performance. If necessary, iterating by adjusting the weighting matrices Q and R can find control gains that minimize a cost function to achieve the desired control performance.

It can be seen that the tracking effect of MPC+LST control is similar to that of LQR control at low speed, but as the vehicle speed increases, the tracking effect of LQR control is not as good as that of MPC+LST control at high speed. From the comparison of lateral displacement errors reflected in Figure 24c, Figure 25c and Figure 26c, it can also be seen that no matter the speed, the lateral displacement error of MPC+LST control is smaller, and the tracking accuracy is higher than that of LQR control. The comparison of heading angle errors reflected in Figure 24d, Figure 25d and Figure 26d shows that the heading angle error of MPC+LST control is significantly smaller in the low- and medium-speed conditions; in the high-speed conditions, although both controllers have larger heading angle errors, the tracking performance of MPC+LST control is better. From the comparison of side slip angle and yaw velocity reflected in Figure 24a,b, Figure 25a,b and Figure 26a,b, it can be seen that the side slip angle and yaw velocity under MPC+LQR control are smaller than those under LQR control, which means the stability of the vehicle in the process of trajectory tracking is better, especially in the high-speed conditions. Overall, MPC+LST control has better tracking accuracy and robustness, and the stability of the vehicle in the control process is also better.

## 8. Conclusions

Aiming at the problem of vehicle dynamics control, lateral controllers and longitudinal controllers are proposed to track the lane change trajectory at high speed. Firstly, the vehicle collision avoidance particle model was built; the requirements of braking collision avoidance, lane change collision avoidance, and braking and lane change combined collision avoidance on the minimum longitudinal distance of the vehicle were analyzed. The results show that compared with braking collision avoidance, the requirement of lane change collision avoidance for longitudinal distance is significantly reduced, and the effect is more obvious with the increase in vehicle speed and the decrease in road adhesion coefficient. Then, the cubic polynomial, quintic polynomial, and hepatic polynomial are analyzed to comprehensively calculate the factors of efficiency, lateral acceleration, and trajectory smoothness. The fifth-degree polynomial is the final reference path. Next, the optimization of the lateral displacement tracking control strategy is to minimize the lateral position deviation, yaw rate tracking deviation, and control increment. The output of the multiobjective lateral MPC is the front wheel steering angle, and the longitudinal speed controller outputs the throttle opening or the brake master cylinder pressure. Finally, the simulation results show that the combined control of the optimized lateral control strategy and the longitudinal speed tracking control strategy can not only achieve more accurate tracking but also ensure that the vehicle will not roll and slip.

Future research directions include: (1) The trajectory tracking controller studied in this paper simplifies the vehicle modeling, especially in the longitudinal speed tracking control. However, existing vehicle systems have strong nonlinearity. Therefore, research on the dynamic control of these aspects can be further intensified. (2) The trajectory planning can be based on the driving motive, which is not only for the self-vehicles but also for other traffic vehicles. (3) More constraints can be considered for the controller to conform to reality.

## Figures and Tables

**Figure 1 sensors-23-05301-f001:**
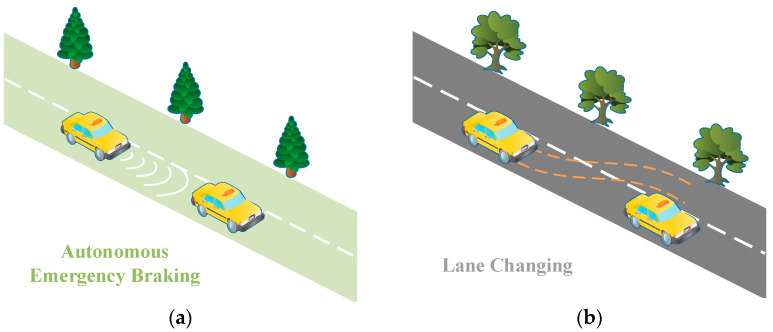
(**a**) Automatic emergency braking to avoid collisions; (**b**) lane changing to avoid collisions.

**Figure 2 sensors-23-05301-f002:**
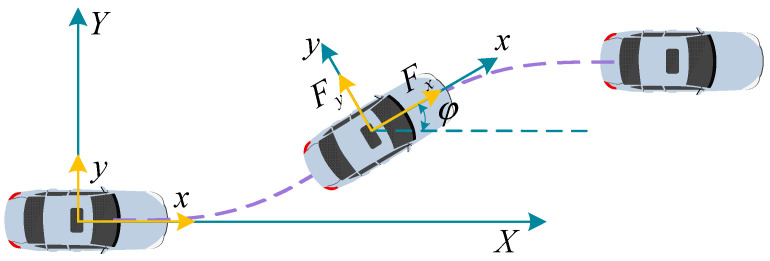
Particle model for vehicle collision avoidance. The purple dotted line shows the vehicle’s driving trajectory. The arrows represent the direction of the coordinate axis or the direction of the force.

**Figure 3 sensors-23-05301-f003:**
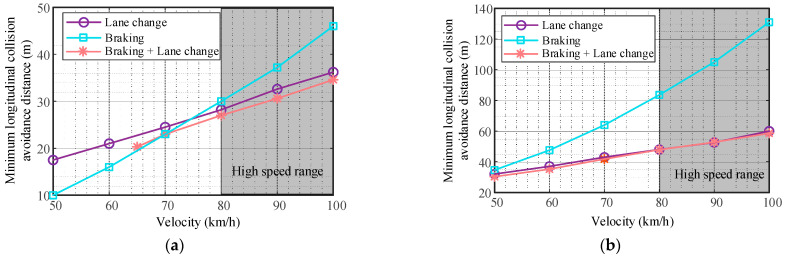
Minimum longitudinal collision avoidance distance. (**a**) (μ=0.85,a=3.5m), (**b**) (μ=0.3,a=3.5m).

**Figure 4 sensors-23-05301-f004:**
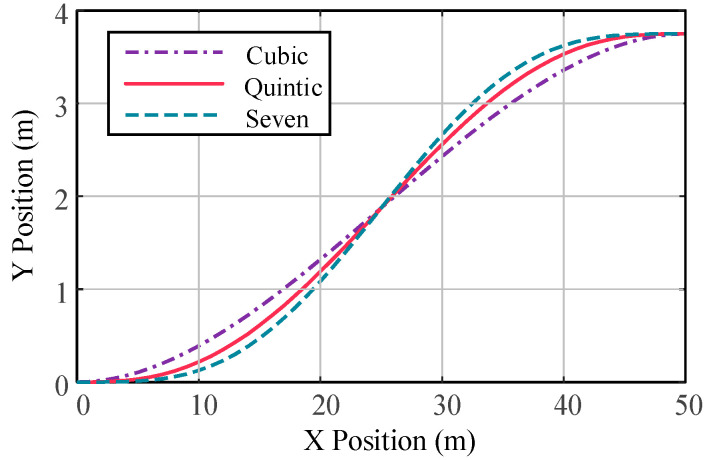
Comparison of different polynomial reference paths.

**Figure 5 sensors-23-05301-f005:**
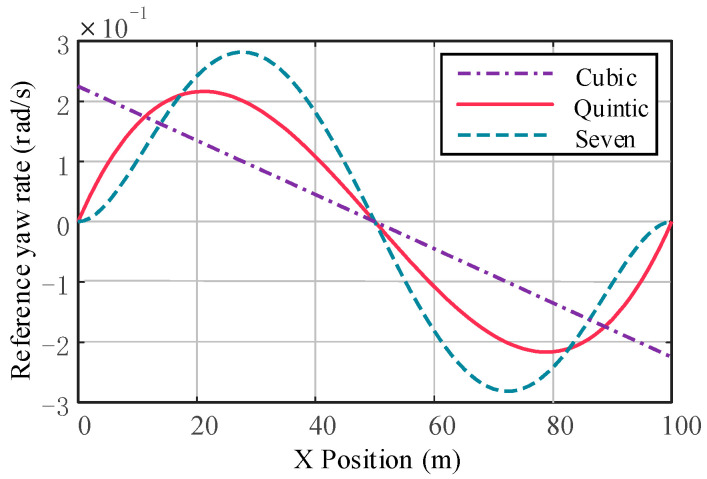
Comparison of the reference yaw rate.

**Figure 6 sensors-23-05301-f006:**
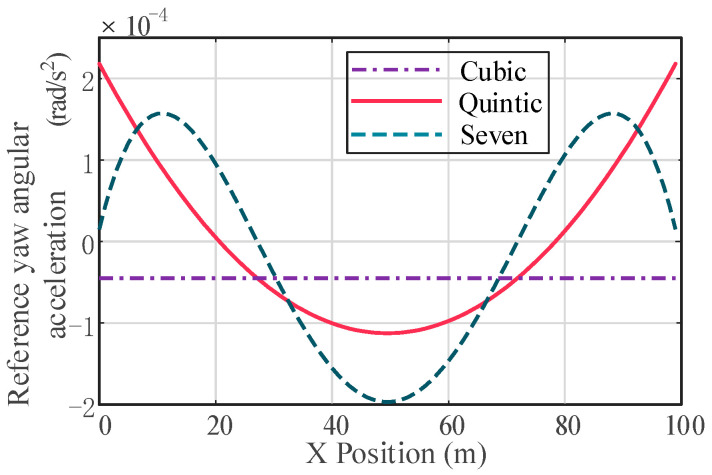
Comparison of the reference yaw angular acceleration.

**Figure 7 sensors-23-05301-f007:**
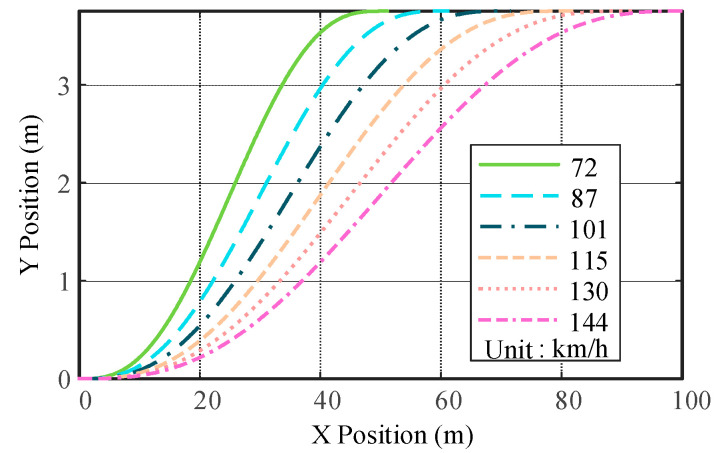
Lateral reference displacement of fifth order at different speeds.

**Figure 8 sensors-23-05301-f008:**
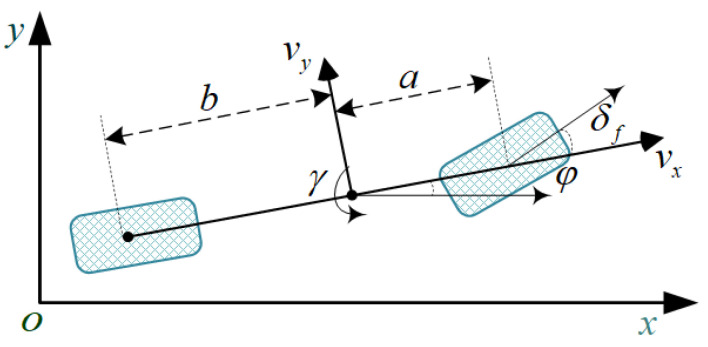
Two-DOF vehicle dynamic model.

**Figure 9 sensors-23-05301-f009:**
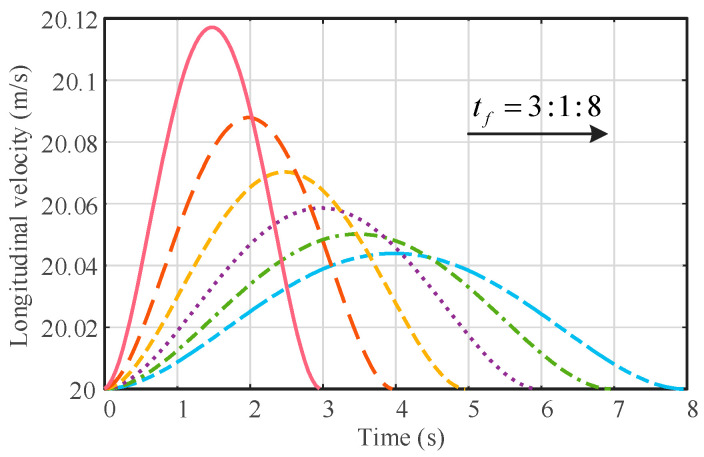
The effect of different lane change times on longitudinal vehicle speeds. The solid or dotted lines of different colors represent the longitudinal speed variation curves at different lane change times; e.g., the peach-colored solid line represents the longitudinal speed variation curve when the lane change time is 3 s.

**Figure 10 sensors-23-05301-f010:**
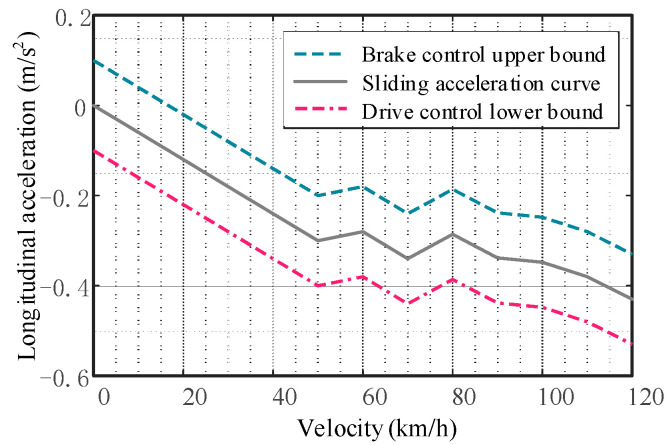
Switching logic.

**Figure 11 sensors-23-05301-f011:**
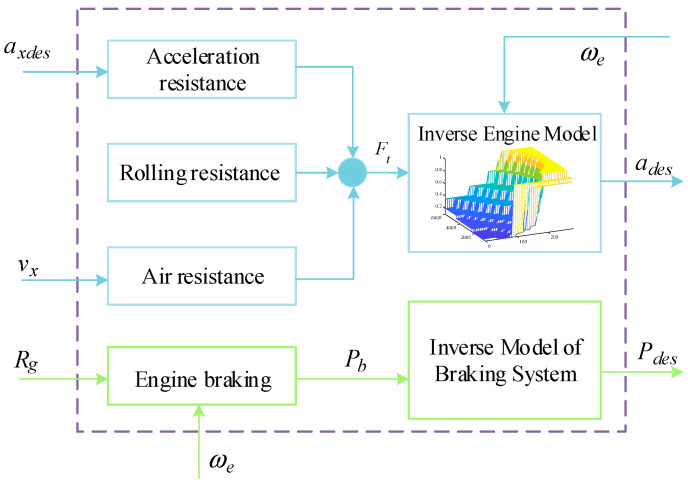
Vehicle inverse longitudinal dynamics model. The sky blue frame represents the inverse engine model and the green yellow frame represents the inverse brake system model.

**Figure 12 sensors-23-05301-f012:**
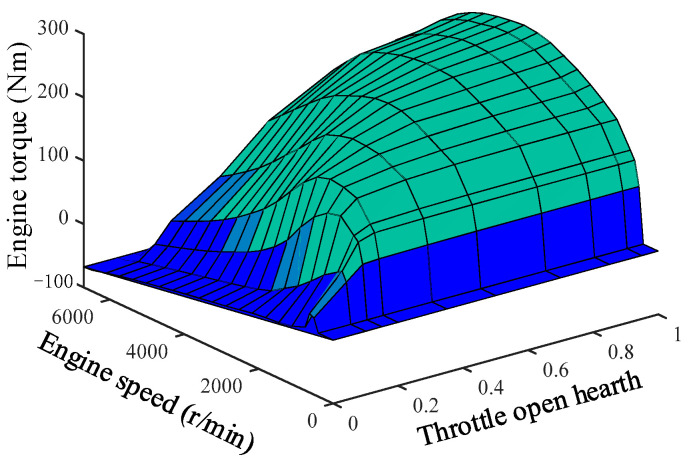
Inverse MAP of engine.

**Figure 13 sensors-23-05301-f013:**
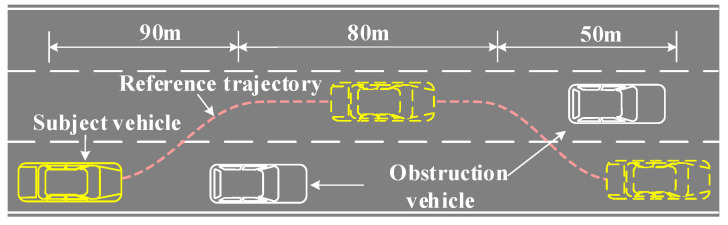
Vehicle collision avoidance scene indication.

**Figure 14 sensors-23-05301-f014:**
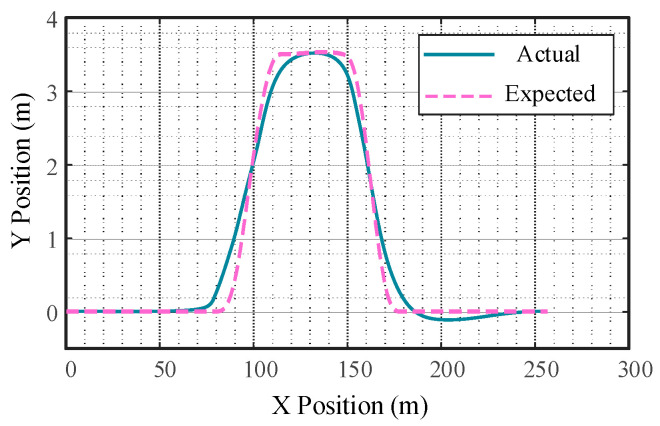
Vehicle lateral-longitudinal displacement.

**Figure 15 sensors-23-05301-f015:**
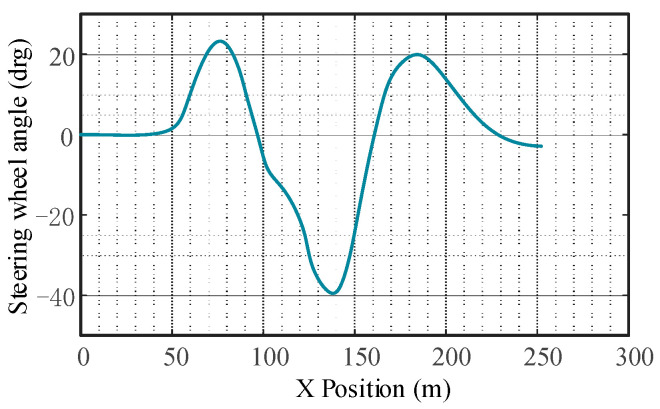
Change in vehicle steering angle.

**Figure 16 sensors-23-05301-f016:**
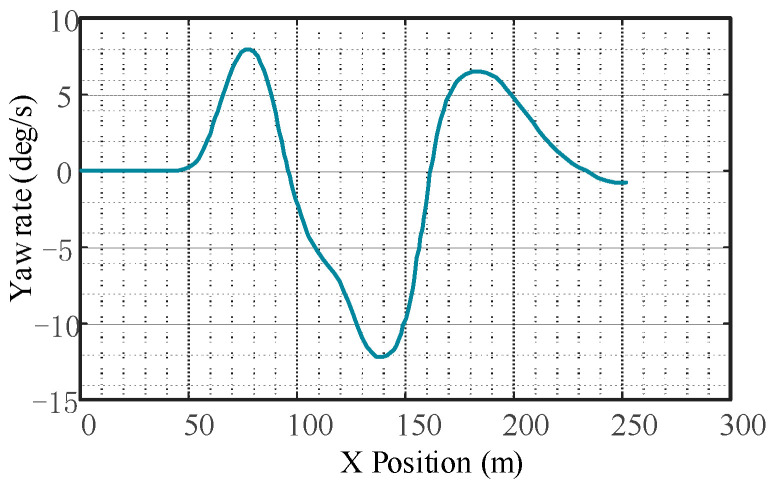
Change in vehicle yaw rate.

**Figure 17 sensors-23-05301-f017:**
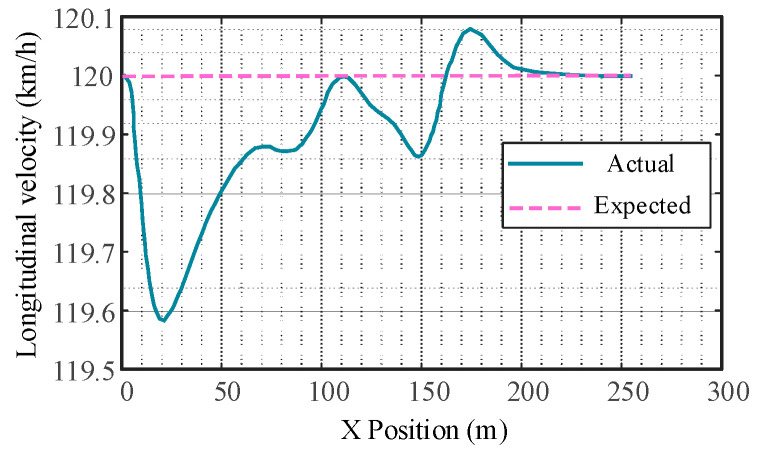
Vehicle longitudinal velocity tracking error.

**Figure 18 sensors-23-05301-f018:**
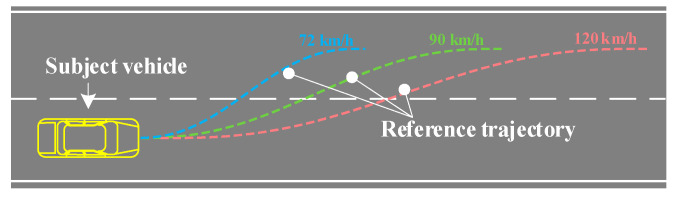
Quintic polynomial reference trajectories at different vehicle speeds.

**Figure 19 sensors-23-05301-f019:**
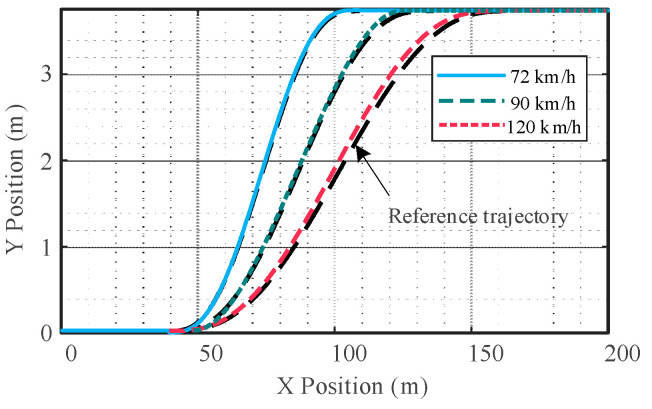
Tracking effect at different speeds.

**Figure 20 sensors-23-05301-f020:**
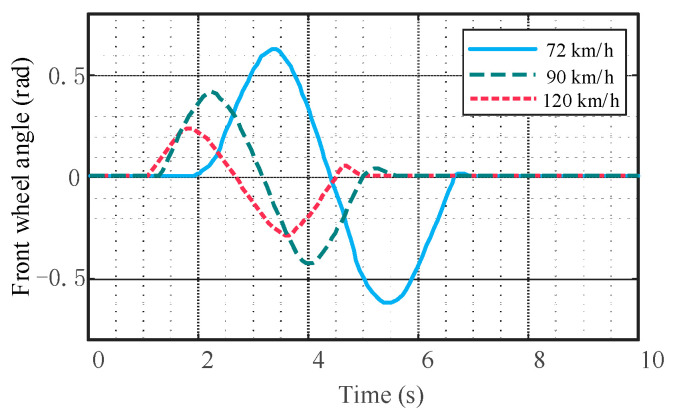
Comparison of vehicle front wheel angle.

**Figure 21 sensors-23-05301-f021:**
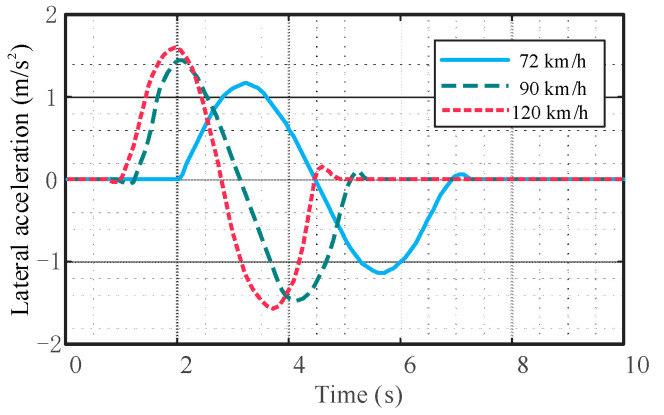
Comparison of vehicle lateral acceleration.

**Figure 22 sensors-23-05301-f022:**
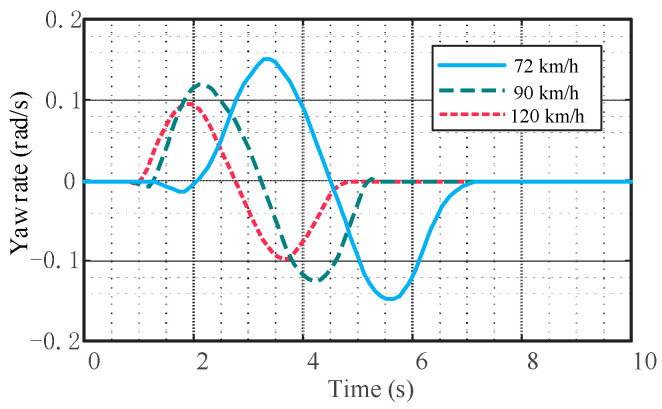
Comparison of vehicle yaw rate.

**Figure 23 sensors-23-05301-f023:**
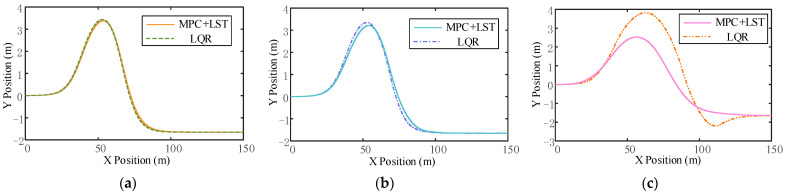
(**a**) Comparison of tracking trajectory effect between MPC+LST control and LQR control (v = 10 m/s); (**b**) comparison of tracking trajectory effect between MPC+LST control and LQR control (v = 20 m/s); (**c**) comparison of tracking trajectory effect between MPC+LST control and LQR control (v = 30 m/s).

**Figure 24 sensors-23-05301-f024:**
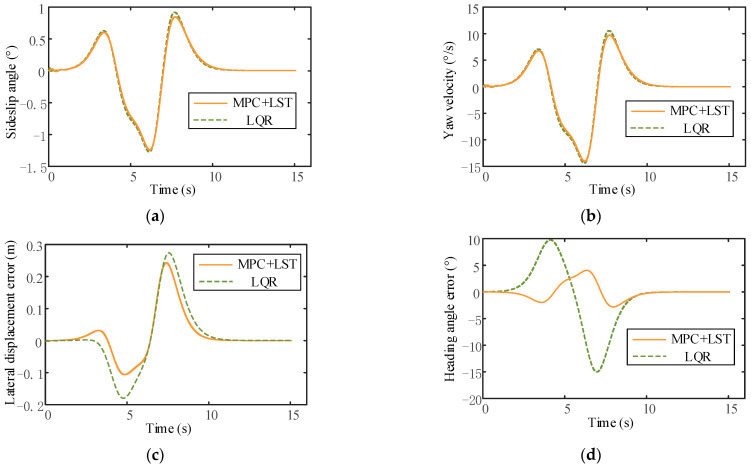
(**a**) Comparison of side slip angle between MPC+LST control and LQR control (v = 10 m/s); (**b**) comparison of yaw velocity between MPC+LST control and LQR control (v = 10 m/s); (**c**) comparison of lateral displacement error between MPC+LST control and LQR control (v = 10 m/s); (**d**) comparison of heading angle between MPC+LST control and LQR control (v = 10 m/s).

**Figure 25 sensors-23-05301-f025:**
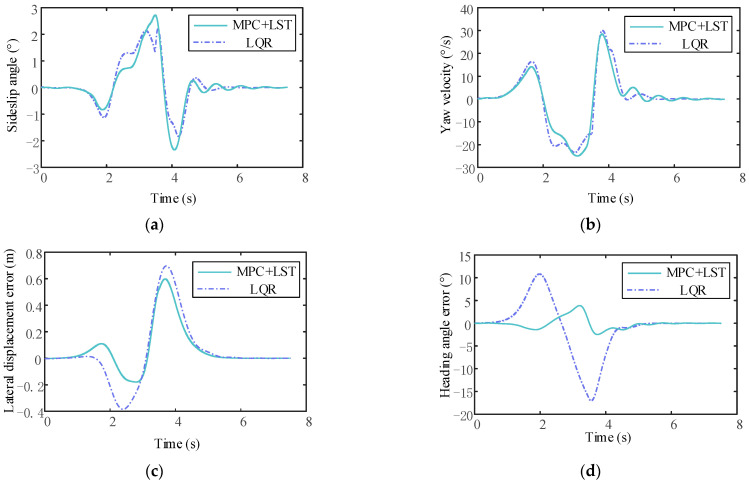
(**a**) Comparison of side slip angle between MPC+LST control and LQR control (v = 20 m/s); (**b**) comparison of yaw velocity between MPC+LST control and LQR control (v = 20 m/s); (**c**) comparison of lateral displacement error between MPC+LST control and LQR control (v = 20 m/s); (**d**) comparison of heading angle between MPC+LST control and LQR control (v = 20 m/s).

**Figure 26 sensors-23-05301-f026:**
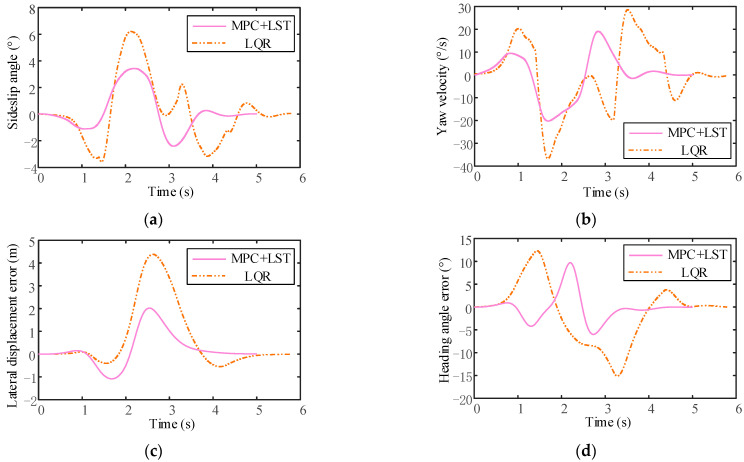
(**a**) Comparison of side slip angle between MPC+LST control and LQR control (v = 30 m/s); (**b**) comparison of yaw velocity between MPC+LST control and LQR control (v = 30 m/s); (**c**) comparison of lateral displacement error between MPC+LST control and LQR control (v = 30 m/s); (**d**) comparison of heading angle between MPC+LST control and LQR control (v = 30 m/s).

**Table 1 sensors-23-05301-t001:** System input and constraint conditions under different collision avoidance modes.

Avoid Collision Mode	Constraint Condition	Input of System
Braking	$Y_{t}=0,v_{Xt}=0,v_{Yt}=0$	$F_{x}\neq 0,F=0$
Lane changing	$Y_{t}=a,v_{Yt}=0$	$F_{x}=0,F_{y}\neq 0$
Braking + lane change	$Y_{t}=a,v_{Yt}=0$	$F_{x}\neq 0,F_{y}\neq 0$

**Table 2 sensors-23-05301-t002:** Some vehicle parameters.

Avoid Collision Model	Constraint Condition	Input of System
Parameter name	value	unit
Sprung mass	1370	kg
Vehicle width	2131	mm
Centroid height	520	mm
Distance from center of mass to front axle	1110	mm
Distance from center of mass to rear axle	1760	mm
Main reduction ratio	4.1	-
Engine power	150	kw
Yaw moment of inertia	250	kg/m^2^
Wheelbase	2886	mm
Tire model	215/55 R17	-
$I_{zz}$	2315	$\mathrm{kg}\cdot m^{2}$

## Data Availability

As the project involves confidentiality, research data is not provided. If readers need research data, please contact the corresponding author.

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
