# Peer review of "Longitudinal and Lateral Control Strategies for Automatic Lane Change to Avoid Collision in Vehicle High-Speed Driving"

_sensors, 2023, doi:10.3390/s23115301_

Round 1

Reviewer 1 Report (Previous Reviewer 3)

An appropriate citation should be included in the manuscript regarding the model described in Eqs.3-7.

The authors should also include a brief description of the LQR tuning method and/or parameters used in their experiments in order for the comparison to be meaningful.

Author Response

Reviewer 2 Report (New Reviewer)

The paper is a revised version for second-round review. There are no major issues in the content and the paper is well organized. Since the paper is about vehicle control, which needs to consider sensor characteristics, vehicle motion states, the paper could be further improved by discussing more related work: automated driving systems data acquisition and processing platform, improved vehicle localization using on-board sensors and vehicle lateral velocity, autonomous vehicle kinematics and dynamics synthesis for sideslip angle estimation based on consensus kalman filter, automated vehicle sideslip angle estimation considering signal measurement characteristic

Author Response

This manuscript is a resubmission of an earlier submission. The following is a list of the peer review reports and author responses from that submission.

Round 1

Reviewer 1 Report

The following points should be considered for improving the quality of the paper:

- Extensive literature review and highlight  the novelty of the paper with repsect to other similar approaches (e.g.  Cavanini, Luca, et al. "LPV-MPC Path Planning for Autonomous Vehicles in Road Junction Scenarios." 2021 IEEE International Intelligent Transportation Systems Conference (ITSC). IEEE, 2021.)

-  include future directions of research 

- review the paper for removing gramatical and typos errors

-

Reviewer 2 Report

This manuscript tentatively discusses control strategies for vehicles traveling at high speeds to avoid collisions, but the control theory and hypothesis testing is difficult to be realized. And the following items that should be carefully treated.

1. As a comparison of two collision avoidance strategies, Figure 1 and Figure 2 should be integrated into one diagram.

2. The detailed deduction of the axial force of the entire vehicle should be added, and it is impossible to obtain it from Equations 1 and 2?

3. The constraints for the two strategies of lane change and braking + lane change in Table 1 are the same. Does this mean that the vehicle does not completely stop when braking? So what is the basis for longitudinal velocity attenuation?

4. The curves in Figures 4 and 5 should be explained carefully in the manuscript. Especially the vehicle information and simulation environment, and detailed calculation steps.

5. In Chapter 2, the collision prevention strategies need to be considered for actual vehicle operation status, surrounding road information, and other vehicle behaviors.

6. The deduction process of equation 10, Equation 13, and Equation 16 should be added in the manuscript.

7. The conclusion in Chapter 3 is also not convincing. The simulation environment should be added.

8. The horizontal tracking control strategy looks more like an optimization method for numerical calculation. And it is impractical to assume that the total vehicle speed is constant when changing lanes. Moreover, neither the driving speed nor the behavior prediction of the front vehicle in the simulation have given the significance of discussing longitudinal acceleration control?

Reviewer 3 Report

Please find my review in the attached pdf.

Reviewer 4 Report

The paper proposes a linear MPC controller for evasive manoeuvres of automated vehicles. The proposed strategy is assessed using a simulation environment. The reviewer’s comments can be found below:

1)      The reviewer recommends using “automated vehicle” instead of “autonomous vehicle”. The SAE J3016 specifies “autonomous” as the deprecated term used for a long time in the robotics and artificial intelligence research communities.

2)      The research contribution compared to the state-of-the-art should be clearly stated.

3)      Figure 8: zero lateral acceleration for cubic representation is confusing. Please specify that it has been derived based on the reference path. Or in the reviewer’s opinion, Figure 8 could be completely removed.

4)      The complexity of the vehicle model used in the controller significantly influences the tracking performance. For more details, the reviewer recommends checking the following paper: Chowdhri, N. et al. Integrated nonlinear model predictive control for automated driving, 2021.

5)      A generic description of MPC can be reduced.

6)      Please provide the information about the controller: prediction horizon, sampling time, etc.

7)      A more detailed analysis (e.g., in the form of sensitivity analysis) regarding the effect of weight selection, prediction horizon, control horizon, and sampling time on performance and computational efficiency should be presented.

8)      Table 2: some information such as Ixx, Iyy, is irrelevant. Tire model should not have any units.

9)      It would be beneficial to add key performance indicators to assess the controller’s performance.

10)   The benchmark comparison should be added.

Round 2

Reviewer 2 Report

The manuscript is not acceptable at its present form as lots of important information is lost.
(1) The detailed deduction of the axial force of the entire vehicle is the basis of the whole control strategy, and it cannot be explained by Equations 1 and 2. The authors should not simplify the model without any deduction.

(2) The authors have not sufficiently explained the information and simulation conditions for Figs 4 and 5.

(3) The collision prevention strategies should consider the actual vehicle operation status, surrounding road information, and other vehicle behaviors.

(4) As lots of constraints have not been considered, the proposed method cannot be used in the practical application. Especially the lack of the theoretical deduction will significantly reduce the novelty of the manuscript.

Reviewer 3 Report

Please find my review in the attached file

Reviewer 4 Report

The reviewer appreciates the authors’ response and correction. However, some previous comments are not sufficiently addressed.

1) There are a lot of studies focused on high-speed trajectory control. The claimed contribution does not align with the state-of-the-art. In addition, the lateral stability of a vehicle cannot be improved using a linear bicycle model.

2) The authors need to clearly define the focus of their study. The nonlinear MPC is used for situations related to nonlinear dynamics. The usage of the linear MPC is valid only for the linear handling range.

3) As mentioned before, a detailed analysis including the corresponding results is needed to argue the selection of design parameters. The authors provided a limited discussion in this regard.

4) Any research publication should include a comparison of the proposed solution with the state-of-the-art. Unfortunately, the authors ignored the previous suggestion referring to the limited time available. In addition, the remarks about stability are is weak. No scenarios are considered in the paper which might lead to an extensive sideslip angle or rollover.
